# Clinical Applications of Combined Immunotherapy Approaches in Gastrointestinal Cancer: A Case-Based Review

**DOI:** 10.3390/vaccines11101545

**Published:** 2023-09-29

**Authors:** Yesim Eralp, Utku Ates

**Affiliations:** 1Maslak Acıbadem Hospital, Acıbadem University, Istanbul 34398, Turkey; 2Biotech4life Tissue and Cell R&D Center, Stembio Cell and Tissue Technologies, Inc., Istanbul 34398, Turkey

**Keywords:** gastrointestinal cancer, immunotherapy, cancer vaccines, dendritic cell vaccine, adoptive cell treatment, cytokine-induced killer cells, immune checkpoint inhibitors, angiogenesis inhibition

## Abstract

Malignant neoplasms arising from the gastrointestinal (GI) tract are among the most common types of cancer with high mortality rates. Despite advances in treatment in a small subgroup harboring targetable mutations, the outcome remains poor, accounting for one in three cancer-related deaths observed globally. As a promising therapeutic option in various tumor types, immunotherapy with immune checkpoint inhibitors has also been evaluated in GI cancer, albeit with limited efficacy except for a small subgroup expressing microsatellite instability. In the quest for more effective treatment options, energetic efforts have been placed to evaluate the role of several immunotherapy approaches comprising of cancer vaccines, adoptive cell therapies and immune checkpoint inhibitors. In this review, we report our experience with a personalized dendritic cell cancer vaccine and cytokine-induced killer cell therapy in three patients with GI cancers and summarize current clinical data on combined immunotherapy strategies.

## 1. Introduction

Despite improvements in screening and treatment, gastrointestinal (GI) tract cancers are a major contributor to the global cancer burden. According to the Globocan statistics, with 3,574,000 new cases, the incidence of esophageal, gastric and colorectal cancers (CRC) represents 19.7% of all cancers diagnosed in 2020, leading to 22.5% of all cancer-related deaths [1]. In the US, with 213,000 new cases and 85,000 deaths projected for 2023, GI cancers rank third in incidence and mortality among all sites diagnosed with cancer. Advances in early detection and novel treatment options have led to a steady decrease in mortality rates over the last decade, reflected by the improved 5 year survival rates for all stages combined for both colorectal (CRC) and gastric cancers. Nevertheless, further research on innovative and effective therapeutic strategies is required to improve the outcomes as GI-related cancer deaths comprise about 13.7% of all cancer deaths and an annual rise in mortality by 1% is noted among young adults with CRC [2,3]. 

With the advent of modern immunotherapy, a new era of cancer therapy has been initialized. Markedly improved survival rates have been achieved with immune checkpoint inhibitors for several cancer types, which had otherwise been considered the most lethal of all cancer types. Programmed death ligand-1 (PD-1) inhibitors, namely nivolumab and pembrolizumab have been approved for the upfront treatment of advanced esophagogastric cancers combined with chemotherapy based on the significantly improved survival rates with hazard ratios (HR) ranging between 0.71 and 0.74 [4,5]. Furthermore, monotherapy with immune checkpoint inhibitors has yielded favorable response rates and prolonged survival rates in microsatellite instable/mismatch repair deficient cancers as compared to standard chemotherapy, earning a rightful designation of “the game-changer treatment” even in heavily pretreated groups [6,7,8,9,10]. Nevertheless, the accumulating data have shown that the majority of patients still succumb to their disease, indicating a requirement for novel approaches to overcome resistance. 

The immunosuppressive microenvironment (TME), inadequate effector immune response and function of cancer-specific cytolytic CD8 (+) T cells, inefficient antigen presentation as well as epigenetic modifications that control immune regulation are major mechanisms that are implicated in acquired resistance to immune-mediated cell killing [11]. Addressing some of these tumor escape pathways, cancer vaccines and the adoptive transfer of T cells combined with chemotherapy, anti-angiogenic agents and radiotherapy have been established as complementary methods to induce an effective anti-cancer immune response [12]. 

In this review, we will discuss the role of immunotherapy in GI tract cancers based on our experience with combined strategies and place it in context with emerging evidence from clinical trials. We hope that this overview will provide insight into comprehensive immunotherapy approaches not only for GI cancers, but for all cancer types where immune-associated cell killing is warranted.

## 2. Cancer Immunology and Immunotherapy 

The innate immune system is hard-wired to generate an immune response by differentiated elements of the myeloid system upon encounter with foreign structures including cancer cells. When tumor cells are identified by the innate immune system, cellular fragments or foreign antigens from these cells are internalized by the antigen-presenting cells (APC) comprised of dendritic cells or macrophages, leading to the orchestration of the functional cascade to prime antigen-specific B and T cell responses [13].

Dendritic cells, a potent member of the mononuclear phagocyte system, play a major role in the activation of innate immunity and the stimulation of naive T cells against tumor-derived neoantigens [14]. Despite the initial discovery by Paul Langerhans in 1868, the term “dendritic cell” was used more than a century later in 1973 to describe cells isolated from murine spleens exhibiting spike-like cytoplasmic extensions. Dendritic cells (DC) originate from CD34 (+) hematopoietic stem cells in the bone marrow to differentiate from common myeloid progenitors into various subsets including conventional, plasmacytoid, inflammatory DCs and Langerhans cells [15,16]. Immature DCs are widely distributed in the blood and stromal tissues to detect and internalize pathogenic or foreign antigens. Uptake of the neoantigen leads to phenotypic and functional maturation of the DCs through a feedback mechanism mediated by the expression of several cytokines in the stromal environment, facilitating mobilization and migration to secondary lymphoid tissues where the processed neoantigen is cross-presented to T and B lymphocytes [15,16,17]. Mature DCs have the capability to bridge both innate and adaptive immune responses by triggering activation of antigen-specific cytotoxic T cell activity, as well as generation of non-specific immune-mediated cell killing by natural killer (NK) cells [18]. 

Although the majority of DCs originate from myeloid precursors, a small fraction may arise from common lymphoid progenitors [19]. Common myeloid progenitors differentiate into macrophage/dendritic cell progenitor cells, which supply the immune system continuously with monocytic APC including DCs. The progenitor DCs produced in the bone marrow migrate to lymphoid and non-lymphoid tissues, where they undergo maturation processes giving rise to functional DCs [20,21]. These “immature” DCs, distributed throughout the body, play a significant role in immune surveillance by detecting and engulfing any foreign antigens through receptor-mediated phagocytosis or macro-pinocytosis [21,22]. Upon maturation, DCs acquire an intensified expression of cell surface receptors, such as major tissue compatibility complex class II (MHC-II) and CD80/CD86 for an efficient antigen presentation as well as improved migratory capacity. Migration through the lymphatic system is signaled by coupling of the chemokine receptor CCR7 with its ligands CCL19 and CCL21 expressed by lymphatic endothelial cells, which is activated in response to increased secretion of the pro-inflammatory cytokine tumor necrosis factor-alpha [23,24]. The antigens presented via the MHC I and II molecules on the cell surface are transferred after binding to the T cell, which takes place in the paracortical section of the lymph node. This transaction stimulates signaling pathways supported by several cytokines in the stromal microenvironment, priming T- and B-cells to induce an antigen-specific immune response [13,18,25]. 

Nevertheless, during tumor progression, cancer cells interact with the host immune system to overcome anti-tumor immunity by switching the tumor microenvironment to an immuno-suppressive or immune-cold phenotype mediated by several cytokines and molecules. The anti-cancer immune response is hindered by the upregulation of various checkpoints and immune suppressive elements of the host immune system, which are activated upon any excessive immune activity against the host. Inhibitory immune checkpoints such as LAG-3, TIGIT and IDO suppress activation of tumor-specific T and B lymphocytes, leading to a shift toward an immunologically unfriendly stroma with excessive infiltration of regulatory T cells (Treg), immature dendritic cells, M2 macrophages and myeloid-derived suppressor cells (MDSC) [26,27,28].

Immunotherapy involves various treatment strategies that aim to restore a dysregulated immune function or to coordinate the generation of an adequate immune-mediated anti-tumor response. The current immunotherapy approaches address several mechanisms that hamper the generation of a sufficient immune response. Monoclonal antibodies targeting the PD-1, PD-L1 and CTLA-4 receptors, which have revolutionized cancer immunotherapy, contribute to the activation of antigen-specific cytotoxic T cells by releasing the brakes on immune cells. As mentioned above, nivolumab and pembrolizumab are PD-1 inhibitors that have been approved for the treatment of gastric and CRC for various indications. Other investigational immunotherapy approaches aim to stimulate the host immune system by cancer vaccines; whereas others involve the administration of ex vivo activated autologous or allogeneic immune cells, such as tumor-associated lymphocytes (TIL), CAR-T cells or engineered NK cells, otherwise referred to as adoptive cell therapy [29,30,31,32,33,34]. Detailed information on these strategies will be provided in context with current evidence in GI immunotherapy in the corresponding sections below.

## 3. Immunotherapy in GI Cancers: Cellular Treatment Systems

### 3.1. Cancer Vaccines: Rationale, Limitations and Potential Combinations

The history of immunization for cancer therapy dates back to 1890, when Sir William Coley treated cancer patients with streptococcal bacterial cultures and reported tumor regressions [35]. Since then, enormous efforts have been placed to design an efficient system, which would stimulate the host immune system to elicit an adequate and specific immune response against cancer cells. Biologic systems that aim to prime T and B lymphocytes by introducing tumor neoantigens via distinct delivery systems are called cancer vaccines. The neoantigens are provided as inactivated whole tumor cells or components of cells that convey specific characteristics, such as nucleic acids or peptides, which lead to the recognition of the cancer cells by effector lymphocytes of the host immune system. These antigens are delivered through several systems including dendritic cells, viral particles, exosomes, or synthetic systems such as lipid and polymer-based nanoparticles. As an active immunotherapy strategy, the goal of treatment with cancer vaccines is not only to induce a cancer-specific response, but also to establish immunologic memory for a sustained and prolonged immune-mediated cell-killing effect [36,37,38]. Despite the enthusiasm generated over immunotherapy in the past century, the clinical efficacy of cancer vaccines has been underwhelming except for Spileucel-T, an autologous tumor cell-based vaccine, which was approved for routine clinical use based on overall survival advantage for patients with castration-resistant prostate cancer and Talimogene Laherperepvec for malignant melanoma based on efficacy to elicit durable tumor responses in remote lesions [39,40,41]. 

#### 3.1.1. Dendritic Cell Vaccines

Human dendritic cell subsets loaded with tumor-associated antigens, referred to as dendritic cell vaccines have emerged as a promising immunotherapeutic tool for the treatment of many cancer types, including colorectal cancer, breast cancer, and hepatocellular carcinoma [42,43,44]. The mainstay of designing dendritic cell vaccines is the development of an effective and durable cytotoxic T-lymphocyte immune response by inducing both humoral and cellular immune responses that, together with the immune capacities of dendritic cells, induce the clonal expansion of T cells [45]. Once activated, dendritic cells have the capability of migrating to the nearest lymph nodes, which is required to initiate immunologic mechanisms leading to the generation of humoral and cellular immune responses against tumors [46,47]. Generally, vaccine production involves isolation of immature dendritic cells or precursors from the blood, followed by ex vivo maturation and activation through co-culturing with cytokine cocktails. The generated mature dendritic cells are then pulsed with autologous tumor cells or neoantigens to form dendritic vaccines which are administered back to the patient [48]. Encouraging preclinical data on the anti-tumor efficacy of dendritic cells packed with tumor lysates led to intensive clinical evaluation of dendritic vaccines by the end of the last century [49]. Subsequent clinical trials utilizing antigen-specific dendritic cells showed partial responses coupled with anti-tumor immune responses in several tumor types including B-cell lymphoma, melanoma and hepatocellular carcinoma [50,51,52,53]. 

#### 3.1.2. Cancer Vaccines and Combined Strategies to Improve Immunogenic Activity in GI Cancers

Nevertheless, cancer vaccines against pre-identified tumor neoantigens in CRC have shown limited efficacy, as the “cold” tumor microenvironment (TME) has precluded the generation of an adequate specific cytotoxic T cell response. Therefore, energetic efforts are being placed to overcome the immunosuppressive TME and turn “cold” tumors into “hot” tumors, where an effective immune-mediated response can be triggered through activation of CD8 (+) T cells, Th1 B cells as well as inhibition of Tregs, tumor-associated macrophages (TAM’s) and proinflammatory signals such as TNF-alpha. Among many strategies, activating cytokine adjuvants used along with vaccines have shown enhanced tumor-specific immune responses. A murine model with CRC, GM-CSF and IL-2 combination with a vaccine targeting WNT as a neoantigen has yielded an augmented T cell response which translated into effective tumor cell killing [54]. Despite discrepant effects of interleukins in the ability to induce both pro-inflammatory and anti-inflammatory responses in the TME, pretreatment with IL-2 has also been shown to improve the efficacy of chemotherapy in a group of metastatic CRC patients by increasing lymphocyte proliferation [55]. Interferon-alpha is another cytokine that plays an important role in generating an immune stimulatory response through upregulating MHC-I molecules on the surface of dendritic cells and in turn improving neoantigen presentation. Furthermore, the anti-angiogenic effect of IFN-alpha helps overcome a significant resistance mechanism by converting a cold TME into an immune-friendly stroma [56,57]. 

As mentioned above, immune checkpoint inhibitors have not shown clinical efficacy in microsatellite stable (MSS) GI tumors through mechanisms that involve increased T cell exhaustion and activation of co-inhibitory signals such as TIM3 and Lag3 [58,59,60]. In fact, combined inhibition of PD-1 and TIM-3 pathways has resulted in enhanced cytotoxic T cell responses to autologous dendritic cell vaccination in CRC patients, addressing an unmet need in this patient population [61]. Chemotherapy has also been shown to activate the immune system by modifying the TME and may have a potential role in converting “cold” into a “hot” TME with a high TIL infiltration [62]. As hypothesized, chemotherapy and CEA vaccine combinations have yielded increased antigen-specific T cell responses and improved clinical responses in earlier clinical trials, providing a strong rationale for integrating cytotoxic agents in vaccination protocols [63,64]. 

Angiogenesis, which plays a significant role in escape mechanisms by supplying nutrients to a growing tumor tissue, is a relevant target in carcinogenesis [65]. Further evidence suggests a possible interplay with angiogenesis and the TME that interferes with immune cell adhesions and migration across the vascular-endothelial junctions, thus preventing infiltration of immune effector cells within the tumor tissue [66]. The hypoxic tumor environment mediated through activation of VEGF and angiopoietin pathways provides an unfriendly stroma not only by inactivation of cytotoxic T cells, but also by promoting expansion of inhibitory components of the immune system including Tregs and TAMs [67]. Earlier observations that have shown an association of bevacizumab use with inhibition of Tregs as well as activation of mature dendritic cell and T cell proliferation lend support to the potential role of anti-angiogenic therapy in generating an immune-friendly TME [68,69]. 

#### 3.1.3. Cancer Vaccines for Esophageal and Gastric Cancer: Clinical Evidence

Earlier clinical trials evaluating the role of dendritic vaccines pulsed with peptides, including Her-2 and MAGE, have shown the feasibility of this approach with a low toxicity profile and tumor-specific T cell responses which translated into minor tumor regressions [70,71,72]. More recent trials focused on combination strategies aiming to improve immune responses as well as clinical efficacy. A peptide vaccine combining URLC10 peptide, a neoantigen frequently expressed in gastric cancer and VEGFR1, an anti-angiogenic epitope was evaluated in a cohort of refractory metastatic gastric cancer patients. Although specific cytotoxic T cell responses were detected in 65% of patients, 30% had disease stabilization as the best response and a poor outcome with a median survival of 3.6 months [73]. In a phase Ib trial, 14 patients with Her-2 overexpressing metastatic gastric cancer were treated with a Her-2 peptide vaccine selected for B cell epitopes combined with a non-toxic form of diphtheria toxin, which is not only intended to prime specific memory T cell responses, but also blocks cellular proliferation through targeting heparin-binding receptors on tumor cells. Out of 11 patients eligible for response assessment, the overall response rate was reported as 54.5%, with one patient who achieved a complete response [74]. 

Several trials evaluated the efficacy of vaccines combined with chemotherapy in the advanced setting and as part of adjuvant chemotherapy in patients who underwent surgery for localized gastric cancer. The GC4 was a multicenter trial that enrolled advanced gastric and gastroesophageal cancer patients investigating the role of chemotherapy and a peptide vaccine from the gastrin sequence conjugated with diphtheria toxin. A group of previously untreated 96 patients received a standard treatment with cisplatin and fluorouracil in combination with the vaccine administered intramuscularly in weeks 1, 5, 9 and 25, yielding confirmed ORR of 30% in the evaluable patients. The median TTP and OS were estimated as 5.4 and 9.2 months, respectively, with a significant improvement in the immune-responders as compared to those who failed to generate circulating anti-G17 antibody titers [75]. A similar approach of dendritic cell vaccination pulsed with WNT and MUC1 peptide given in combination with salvage chemotherapy was evaluated in 20 patients with gastric cancer. The treatment was feasible and a trend for prolonged survival was observed in patients in whom a higher percentage of effector T cells could be induced [76]. As an adjuvant strategy, a multi-epitope vaccine combining neoantigens with angiogenic peptides was evaluated in 14 patients who received adjuvant chemotherapy for stage III resected gastric cancer. The combined approach was found feasible, but the outcomes were not reported [77]. 

Immune checkpoint inhibitors revolutionized immunotherapy for the treatment of numerous cancer types. From a mechanistic point of view, these agents are hypothesized to boost cancer-specific cytotoxic T cell responses by releasing the brakes on the immune system. This strategy was explored in a patient with metastatic gastric cancer as part of an ongoing clinical trial investigating a dendritic cell vaccine pulsed with eight tumor-associated neoantigens. The patient received four vaccines after lymphodepletion with cyclophosphamide followed by boosters given in combination with nivolumab every two weeks starting from day 65, leading to a rapid tumor response despite progression over the first two months of single-agent vaccine therapy. The patient displayed a complete response which was ongoing after 25 months [78]. Ongoing and completed clinical trials with vaccines in upper GI malignancies are listed in Table 1. 

#### 3.1.4. Cancer Vaccines for Colorectal Cancer: Clinical Evidence

As a universal biomarker, carcinoembryonic antigen (CEA) is a frequently utilized tumor antigen in CRC vaccine trials. Early phase I trials evaluating the role of DC vaccines pulsed with CEA have demonstrated potent T cell responses in a small cohort of CRC patients with liver metastases. Correlations with outcomes could not be analyzed due to the small sample size [79]. Another study investigating the role of an engineered adenovirus construct encoding a modified HLA-restricted CEA included 32 patients with CRC who had progressed following a median of three previous lines of standard chemotherapy. These patients who received three subcutaneous injections every three weeks were found to generate polyfunctional CD 8 (+) cytotoxic T cells and exhibited an encouraging 12-month survival rate of 48% [80]. Similarly, a MUC-1 peptide vaccine given as three subcutaneous injections with a booster dose at week 52 was shown to generate weakly increased memory cell immune responses in a cohort of 37 patients with CR adenomas, suggesting possible efficacy as a preventive strategy [81]. A recombinant vaccinia virus-based vaccine encoding multiple neoantigens including CEA, MUC-1, ICAM and LFA was evaluated in a randomized study including CRC patients who had undergone resections for liver or lung metastases. Seventy-six patients were enrolled in the trial, which aimed to investigate the role of two different vaccine constructs in a minimal residual disease setting. Translational analysis showed a high ratio of CEA-specific T cell responses. Although there was no difference between the two groups, the median RFS and OS rates reported as 25.7 and 44.1 months, respectively, were higher than contemporary unvaccinated controls [82]. A recent study evaluated the role of an immunomodulatory vaccination with arginase-I (ARG-I) peptide, which targets the immunosuppressive tumor microenvironment (TME) in a small cohort with multiple tumor types including CRC. Nevertheless, despite generation of a specific T and B cell immune response, there were no objective responses attained with the new approach [83]. 

Further clinical trials integrated innovative immunostimulatory strategies in vaccination schedules to improve response rates. One of these, which included 53 advanced cancer patients comprising 38 patients with CRC evaluated the efficacy of a Ras-peptide vaccine. Patients received 3 doses of the vaccine every 3–5 weeks with GM-CSF or IL-2 as immunostimulatory adjuvants. The study demonstrated that almost all patients who received GM-CSF developed an immune response by the ELISPOT assay. An unplanned subgroup analysis of CRC patients revealed an encouraging OS of 14.2 months, which compared favorably to historical controls who had a median OS of 12.9 months, despite the shorter median PFS of 3.5 months in the vaccinated cohort. This interesting observation lends support to the generally accepted notion that response to immunotherapy and overall survival are not correlated [84]. 

Numerous clinical studies have utilized chemotherapy as an immune modulatory approach to improve the efficacy of cancer vaccines. Barve et al. [85] have reported two cases who had been treated with an autologous tumor vaccine given in conjunction with the FOLFOX regimen following resection of liver metastases. The vaccine consisted of surgically resected autologous tumor samples electroporated with a plasmid encoding GM-CSF and a bifunctional siRNA targeting immunosuppressant cytokines. Patients received subcutaneous two injections every two weeks followed by maintenance every four weeks until the product was exhausted along with six cycles of chemotherapy, which resulted in a disease-free survival extending beyond eight years. A similar approach was evaluated in a phase I trial that evaluated the role of a peptide vaccine containing seven tumor neoantigens combined with chemotherapy as an add-on maintenance strategy in the minimal residual disease setting. In addition to sustained tumor-specific T cell responses, a confirmed ORR of 27.5% and a clinical benefit rate of 63.6% was achieved with the combination in a cohort of 11 patients with metastatic CRC [86]. 

Emerging clinical evidence suggests a possible role for immune checkpoint inhibitors in priming T cell responses in an otherwise immune-suppressed TME, such as microsatellite stable CRC. A phase II clinical trial evaluated the role of GVAX, an allogeneic, whole-cell cancer vaccine given in combination with pembrolizumab and low-dose cyclophosphamide in patients who had progressed after at least two prior lines of chemotherapy for mismatch repair proficient metastatic CRC. Unfortunately, the trial was terminated due to futility after the enrolment of 17 patients, in whom no objective responses were observed despite a substantial biochemical response [87]. Similarly, a combination of avelumab and an autologous dendritic cell vaccine yielded no objective responses in a small phase II study comprising of a heavily pretreated cohort with metastatic CRC [88]. Nevertheless, a phase I trial investigating the role of a combined viral-based multiepitope vaccine comprising of a chimpanzee adenovirus and Venezuelan equine encephalitis virus-based vectors has been shown to elicit long-term memory immune responses in 14 patients comprising of 7 MSS metastatic CRC and 8 patients with metastatic GC. In the expanded phase 9 patients (3 GC, 6 CRC) received subcutaneous low-dose anti-CTLA-4 monoclonal antibody (ipilimumab) or PD-1 therapy with nivolumab along with vaccination. In this cohort, 1 patient with GC had a complete response and was alive after 470 days. In the whole cohort, the majority of patients had a stable response. For patients with MSS-CRC, 3 of 7 patients remained alive with an OS of 42% at 12 months, suggesting that some patients may derive durable clinical benefit despite the lack of a radiologic response by imaging [89].

Angiogenic inhibition with bevacizumab has recently been explored as a distinct approach to overcome the immune suppressive environment in a phase II randomized trial. Patients with untreated MSS metastatic CRC were randomized to FOLFOX plus bevacizumab as standard treatment or the same regimen combined with avelumab and an adenovirus-based vaccine encoding CEA. The trial was terminated early after an unplanned futility analysis conducted with eight patients, showing no difference in PFS [90]. 

Despite the capability to generate robust cancer-specific T cell responses, the inadequate clinical efficacy of cancer vaccines combined with several immunomodulatory strategies highlights the requirement for the identification of subgroups who benefit from these therapies, as well as innovative approaches that would boost immunogenic anti-cancer responses. Ongoing and completed clinical trials with vaccines in CRC are listed in Table 2. 

### 3.2. Adoptive Cell Therapy: Cytokine-Induced Killer Cells as an Immunotherapy Approach in GI Cancers

Adoptive cell therapy (ACT) is a type of personalized cellular immunotherapy, that involves isolation of immune cells from the blood and re-administration following ex vivo expansion. Initial evidence regarding the efficacy of ACT has been generated from earlier melanoma trials, which have demonstrated response rates ranging between 49–72%, and long-term survival achieved in a fraction of patients [91]. Cytokine-induced killer cells (CIK) represent a fraction of expanded lymphocytes that are comprised of CD3+CD56+ natural killer cells (NK), which are capable of MHC-unrestricted immune cell killing, as well as CD3+CD56-T cells. Although NK cells are the major effector cells, the anti-tumor efficacy of CIK cells is enhanced through the induction of cytokines that upregulate Th1 type immune response and promote migration to the tumor site following infusion. The main advantage of CIK cells is their potential efficacy against a wide array of tumors, MHC-unrestricted activity and less demanding technical requirements, which have led to the widespread use of this promising strategy as an adoptive immunotherapy tool in several tumor types [92,93]. 

Numerous clinical trials have evaluated the feasibility and efficacy of CIK cell therapy in GI cancers. One of the earlier studies randomized 60 patients with metastatic CRC to standard chemotherapy versus 1–4 cycles of CIK cycles in combination with 6 cycles of FOLFOX chemotherapy. The majority of these patients had undergone surgical resection of metastatic sites. The investigators reported a significant improvement in PFS (25.8 vs. 12 months; *p*: 0.01) and OS (41.3 vs. 30.8 months; *p*: 0.037) in the combined treatment group [94]. A phase II clinical study including 33 patients with stage IV GI cancers evaluated the role of combined CIK and CAPOX chemotherapy in the first-line setting. Two CIK infusions were administered on days 14 and 16 in the first couple of chemotherapy cycles in the first group, whereas the second group of 17 patients received chemotherapy only. Numerically longer PFS and OS rates were achieved with the combination despite failure to show a significant difference (PFS: 5.6 vs. 3.8 months, *p*: 0.06; OS: 13.9 vs. 11 months in combined versus chemotherapy arms, respectively). Although the CD8 (+) ratio was not different, patients in the combined arm showed an improved humoral immune response with increased NK, and CD4 (+) helper T cells [95]. In a prospective phase II trial by Zhao et al. [96] 122 patients were randomized to autologous CIK cells combined with chemotherapy or to chemotherapy alone. The chemotherapy arm consisted of the FOLFOX regimen given for 12 cycles every two weeks and CIK cells were administered on days 15 and 16 of each cycle. As often encountered in immunotherapy trials, with the combined treatment a significant OS (36 vs. 16 months, *p* < 0.001) was achieved, despite a numerically longer PFS not reaching significance levels (16 vs. 10 months, *p*: 0.07). 

The impact of CIK cell-based immunotherapy was also investigated by several trials in the adjuvant setting following resection for early-stage CRC. In a retrospective analysis comprised of 96 patients who received at least one CIK infusion during adjuvant chemotherapy, a significant improvement in DFS was achieved with a HR:0.28 (*p*: 0.034) [97]. A small prospective randomized phase II study enrolling 46 patients reported improved quality of life and longer OS with the combined CIK and adjuvant FOLFOX regimen (41.9 months) as compared to those who received only chemotherapy (33.8 months) [98]. A retrospective study evaluated the role of combined CIK treatment and chemotherapy in a cohort of 60 patients with a control arm comprising of 62 patients who received chemotherapy. Despite a higher ratio of stage III patients in the CIK arm, a significantly higher 5-year DFS (70.7% vs. 48.3%, *p*: 0.0024) and OS rate (88.7% vs. 72.4%; *p*: 0.008) were achieved with the combination arm as compared to controls [99]. 

The encouraging efficacy of adoptive immunotherapy with CIK cells in CRC, either given as a single treatment or in combination with DC was confirmed in a meta-analysis, which included data from 70 studies involving 6743 patients with all stages, of whom 66.7% had stage IV disease. Twenty-five studies included CIK as the only investigational procedure. A substantial difference in PFS (HR:0.63, *p* < 0.00001) and OS (HR:0.57, *p* < 0.00001) was observed with CIK cell immunotherapy as compared to non-immunotherapy treatment [100]. 

### 3.3. Combined İmmunotherapy Approaches: Rationale and Preclinical Evidence

Favorable outcomes achieved with CIK cell therapy led to increased efforts to improve the efficacy of combined approaches. There exists strong preclinical evidence on the synergistic efficacy of pro-inflammatory cytokines such as interleukins [101,102,103,104] and interferon [105] as well as cytotoxic agents [106], novel small molecule inhibitors [107,108,109], immune checkpoint inhibitors [110,111,112], monoclonal antibodies [113,114] and dendritic cell [115,116,117] or viral-based vaccines [118,119,120] combined with adoptive cell therapy. Several stromal or intracellular mechanisms implicated in synergistic activity have been identified. As mentioned above cytokines and cytotoxic agents improve immune-mediated cell killing through TME modulation not only by decreasing the ratio of Tregs, but also by induction of a memory effector cell response [102,104] or direct anti-tumor effect by blockade of cytokine-receptors [105]. Co-incubation with dendritic cells has yielded a higher percentage of CD3, CD4 (+) helper T and CD8 (+) T cell accumulation in cell cultures, resulting in improved anti-proliferative and cytotoxic effects not only through induction of immune-mediated cell killing, but also direct activation of CIK cells through CCR5 signaling [115,116]. Small molecule inhibitors that interfere with immunosuppressive pathways such as angiogenesis or beta-catenin-mediated CXCR3 chemokines have also been shown to enhance effector T and NK cell infiltration within the tumor stroma, thus inducing a friendly T-cell inflamed TME [107,109]. As a complementary strategy, immune checkpoint inhibitors augment the anti-tumor efficacy of CIK cells by neutralizing PD-1 and PD-L1 signaling upregulated on both tumor and CIK cells upon engagement [110]. Recent advances in technology have led to the development of novel gene therapies delivered through viral constructs which directly target cancer cells and exert cytolytic activity [118,119,120]. 

### 3.4. Clinical Applications of Combined Cellular İmmunotherapy in GI Cancer: Dendritic Cell Vaccines and Cytokine-İnduced Killer Cells

Based on preclinical evidence showing synergistic interaction between dendritic and CIK cells, numerous trials have investigated the role of combined immunotherapy with DC vaccines and CIK cells. In general, these trials have confirmed preclinical observations for GI cancers with enhanced humoral immune responses translating into significantly improved ORR and survival rates [121,122,123,124,125,126]. A recent meta-analysis including 70 trials has reported significant PFS (HR: 0.55, *p* < 0.00001) and OS (HR:0.61, *p* < 0.00001) advantage with DC/CIK therapy in CRC for all stages. Although direct comparison with single CIK therapy is not given, benefit rates in outcomes are numerically higher for the DC/CIK combination, providing clinical evidence of the synergistic activity [100]. 

## 4. Case Reports

### 4.1. Case 1

The sixty-six-year-old male patient was admitted to the clinic with epigastric pain, fatigue and exertional dyspnea. Laboratory findings revealed iron deficiency anemia, for which he was referred to the endoscopy unit for further evaluation. A subsequent gastroscopy showed an erosive mass measuring 3 cm × 4 cm. located in the antrum, whereas a colonoscopy denied any pathologic findings. A biopsy obtained from the antral mass confirmed the diagnosis as poorly cohesive adenocarcinoma, consistent with gastric primary. Following the completion of imaging workup which showed multiple regional enlarged lymph nodes without any systemic metastatic involvement, he was recommended to undergo perioperative platin-based chemotherapy followed by surgery. Nevertheless, he refused chemotherapy and opted for upfront surgery. Subsequently, he underwent distal subtotal gastrectomy on 21 August 2017. Pathologic evaluation revealed a poorly cohesive, signet-cell ring adenocarcinoma in the antral region which invaded the serosa as well as the lesser momentum and 32 out of 75 lymph nodes involved, consistent with T4N3M0 gastric cancer. Further analysis by FISH revealed Her2 amplification. At that stage, we discussed the risks and benefits of adjuvant chemotherapy, which he refused and opted for personalized cellular immunotherapy. Following regulatory approval based on encouraging evidence from a phase II prospective study combining adjuvant chemotherapy with a cancer vaccine and cytokine-induced killer cells in patients undergoing surgery with gastric cancer, he underwent a combined immunotherapy program consisting of dendritic cells pulsed with autologous tumor cells and cytokine-induced killer cell infusions intercalated with PD-1 inhibition, and metronomic oral cyclophosphamide for 6 months. The adoptive cell therapy was supported with an in vivo cellular expansion protocol including rh-interleukin-2 and interferon-alpha injections. The detailed treatment protocol is depicted in Figure 1. 

He received four DC/CIK infusions between 10 October 2017 and 20 February 2018, which was lower than the planned schedule due to side effects related to IL-2 injections and limited cellular product yield. In vivo expansion was generally provided by IFN-alpha as IL-2 had to be discontinued due to severe fever, chills, and arthralgias. Starting from the initiation of the treatment protocol, he was routinely screened with imaging studies every twelve weeks and remained disease-free until November 2018, when he started complaining of weight loss. A subsequent endoscopic evaluation revealed recurrent tumoral lesions at the distal anastomosis level extending throughout the colonic mucosa. Following confirmation of Her2 overexpressing adenocarcinoma consistent with relapsed gastric cancer, systemic chemotherapy with trastuzumab was initiated. Unfortunately, after failing several lines of chemotherapy combined with Her-2 inhibition, he was deceased in November 2019 with leptomeningeal involvement. 

### 4.2. Case 2

A 39-year-old male patient was admitted to our clinic in February 2019 with progressive metastatic right colon cancer. His diagnosis dated back to March 2017, when he underwent a right hemicolectomy for a tumor in the caecum. The pathologic evaluation was consistent with a T3N1M0 KRAS mutant, grade 2 mucinous adenocarcinoma. Following adjuvant chemotherapy with the FOLFOX regimen for 3 months, he was followed up until August 2017, when recurrence in the abdominal cavity and liver metastases were noted. He then underwent several cycles of perioperative chemotherapy consisting of oxaliplatin and capecitabine, followed by cytoreductive surgery with HIPEC, as well as radioablative treatment (RFA) to the liver metastases until November 2017. In January 2018 progression in the liver lesions was identified and he received combination chemotherapy with oxaliplatin-irinotecan-fluorouracil and bevacizumab with consolidative RFA that was completed in March 2018. Nevertheless, due to rapid and subsequent recurrences, he underwent a second cytoreductive surgery with HIPEC in May 2018 and a colon resection in January 2019. A comprehensive genomic evaluation of the resected specimen revealed pathogenic mutations in KRAS G12D, TP-53 and amplification of CCND2, FGF23, FGF6 genes. He was then referred to our clinic for a second opinion on personalized immunotherapy strategies. Imaging studies obtained after his admission failed to show any recurrent lesions, consistent with minimal residual disease. After a thorough discussion of further treatment options, he decided to pursue a combined immunotherapy approach consisting of a personalized DC vaccine, CIK adoptive cell therapy, PD-1 inhibition and bevacizumab based on favorable responses reported by several phase II clinical trials. Following regulatory approval, he underwent apheresis by which cellular products were manufactured under GMP conditions. A detailed treatment plan is provided in Figure 2. He received this protocol between 15 May 2019 and 22 August 2019. On the day of the first DC/CIK administration, he experienced a grade 3 infusion reaction with fever, chills and hypotension which required hospitalization. The incident was managed with hydration as well as intravenous medications and did not recur on subsequent therapies. He generally tolerated the protocol well with minimal temporary fatigue throughout the treatment. Unfortunately, an abdominal MRI obtained in September 2019 revealed two implants consistent with peritoneal recurrence. He then went back to his primary treating center for continuation of systemic therapy. In February 2020, he reported being enrolled in a clinical trial with investigational agents which proved to be unresponsive and had been under evaluation for additional systemic therapy. He is still alive with disease progression as of July 2023. 

### 4.3. Case 3

A 47-year-old male patient was referred to our clinic for a second opinion on personalized immunotherapy options as consolidation treatment after induction chemotherapy and surgery for left-sided colon cancer in addition to chemoembolization for liver metastases. His initial diagnosis dated back to March 2018, when a tumor located in the rectosigmoid colon with synchronous multiple liver metastases was detected. After a stable response to three months of systemic chemotherapy with oxaliplatin, irinotecan and fluorouracil combined with cetuximab, he underwent radiofrequency ablation to several liver metastases on 29 May 2018, followed by a low anterior resection procedure with resection of residual liver metastases on 25 June 2018. The pathologic evaluation revealed a grade 2 adenocarcinoma with minimal response to cytotoxic chemotherapy. With no proven evidence for further chemotherapy, we discussed the potential benefits of a combination regimen with DC/CIK infusions administered in conjunction with programmed death-1 (PD-1) and vascular endothelial growth factor (VEGF) inhibitors, based on encouraging outcomes with personalized cellular immunotherapy strategies in the minimal residual disease setting. Following regulatory approval, he underwent apheresis by which cellular products were manufactured under GMP conditions. A detailed treatment plan is provided in Figure 3. 

He received this protocol between 15 February 2019 and 30 April 2019, followed by nine booster DC/CIK infusions until 13 January 2021. He generally tolerated the protocol well, with no major toxicity throughout the treatment. Follow-up imaging studies failed to show any recurrence until 4 July 2022, when a nodular lesion in the right thoracic wall and an FDG-avid lymph node in the right hilar station were detected in a PET-CT scan. He then underwent resection of the thoracic lesion, which confirmed the diagnosis as metastatic adenocarcinoma, consistent with colorectal primary. A comprehensive genomic evaluation revealed pathogenic mutations in NRAS, TP-53, SMAD4, APC and ZFHX3 genes, and no PD-L1 expression. The tumor was microsatellite stable and had a low tumor mutation burden of 3.7 m/MB. He then underwent 6 cycles of systemic chemotherapy with the FOLFOX and bevacizumab combination between June and December 2022, which resulted in a partial regression of the hilar lymph node and no residual disease elsewhere. At this stage, we discussed further therapy options and based on his favorable experience with previous consolidative immunotherapy resulting in a sustained progression-free interval, we decided to pursue a similar immunotherapy strategy following irradiation of the residual lymph node basin. Based on the previous protocol, we initiated the treatment with the conditioning regimen on 4 April 2023, and he received this protocol until the last bevacizumab infusion which was administered on 12 July 2023. There were no major side effects encountered during the immunotherapy period except a prolonged grade 1 diarrhea accounted for by pembrolizumab and managed by symptomatic medications. Response assessment by a PET-CT scan obtained on 24 July 2023 failed to show any foci of active lesions suspicious of residual disease. He is planned to receive booster vaccinations every 2–3 months until the vaccine product is exhausted. Please refer to Figure 3 for details on the final immunotherapy protocol. 

### 4.4. Methods

All cellular therapy products were produced under GMP conditions in an approved facility. Dendritic cells and CIK cells were generated from peripheral blood mononuclear cells expanded ex vivo via incubation with cytokines including IL-1, IFN-alpha, TNF-alpha, and IFN-gamma. Activated mature dendritic cells were further incubated with autologous cancer cells to produce a personalized DC vaccine for each patient. Generally, seven days of culture was sufficient to produce the vaccine, whereas 3–4 weeks were required to produce a sufficient amount of CD3+/CD56+ CIK cells to ensure optimal activity. Although the number of viable cells differed for each apheresis procedure, the ultimate aim was to administer 1–10 × 10^9^ cells at each administration. The dendritic cell vaccine pulsed with autologous tumor cells was split in aliquotes and administered subcutaneously in all four limbs. Frozen CD3+/CD56+ CIK cells suspended in isotonic saline solution were thawed and administered as an IV infusion over 10–20 min.

All subjects gave informed consent for inclusion before participating in the study. The investigational study was conducted in accordance with the Declaration of Helsinki, and individual approvals were obtained by the Ethics Committee of Gayrettepe Florence Nightingale Hospital, Istanbul, Turkey and subsequently from the Ministry of Health, Health Services General Directorate, Department of Tissue, Organ Transplant and Dialysis (ID codes: 02.11.2017/56733164/203; 27.03.2019; no:216/56733164/203; 19.01.2023/E-56733164-203-210725859). 

## 5. Future Prospects and Conclusions

Accumulating evidence suggests a possible beneficial role for active immunotherapy with cancer vaccines and adoptive cell therapy with CIK cells in combination with several immunomodulatory agents for GI cancers. Nevertheless, most of the trials evaluating these strategies suffer from a lack of standardization and imbalances in patient characteristics, which limit our ability to reach a definite conclusion on the optimal clinical use of cellular immunotherapy. Furthermore, both tumor and patient-related factors cause temporal immunologic plasticity over the evolution of disease progression through aberrant genomic signaling and TME modulation which complicate the process even further. Given these constraints, our preliminary experience with cellular immunotherapy showing a lack of efficacy in two out of three patients despite the addition of VEGF and PD-1 inhibition is in line with previous reports highlighting the difficulties in achieving an immune-mediated response in an otherwise “cold” tumor microenvironment seen in GI cancers. Nevertheless, in the patient with de novo metastatic CRC who had been refractory to initial chemotherapy, the progression-free interval of 18 months after completion of immunotherapy is encouraging and may be considered as a signal of activity. However, the major limitation in our protocol that should be noted is the lack of predictive biomarkers to select patients who would benefit from the novel immunotherapy approaches as well as to monitor the humoral immunologic response generated throughout the treatment and follow-up period. There exists translational evidence from experimental models and clinical studies showing correlations of response with increased levels of antigen-specific T cells, activated immune cells such as NK cells with high activating receptor p56 expression and novel immune signatures [60,90]. Nevertheless, contradictory findings have been reported by several other trials investigating similar combined approaches, which have failed to identify predictive or prognostic correlations with humoral immune markers [82,88]. Unfortunately, despite extensive efforts from preclinical and clinical studies, there has not been any proven predictive factor for immunotherapy except microsatellite instability. It’s worth mentioning that the two patients with CRC reported here were microsatellite stable. One patient had been evaluated for PD-L1 expression, which was found to be negative. Given the lack of reliable correlates and the small sample size, we did not plan to assess investigational biomarkers in our patient group. 

Cancer immunotherapy is a rapidly evolving field. Advances in technology have led to the development of engineered cellular systems or synthetic nanotechnology-based materials that tap into the immune system to activate cytotoxic and memory-effector response against cancer cells. Innovative delivery systems such as non-replicating viral particles, nanoliposomes, exosomes or polymeric nanoparticles loaded with immune checkpoint inhibitors, siRNA, or tumor antigens have been shown to hold great potential as an immunotherapeutic by inducing potent anti-tumor response as well as modulating an immunologically “cold” TME as demonstrated by preclinical studies [89,127,128,129,130]. Furthermore, engineered dendritic cells expressing CD40 bispecific antibodies showing anti-cancer activity through enhanced DC and T cell activity, hold promise as vaccines against several cancers [131]. 

Oncolytic viruses which are specifically designed to selectively replicate in cancer cells, possess a potential for dual anti-tumor action by a direct cytolytic effect as well as activation of the stroma to induce an immunologic effector cell response. Engineered oncolytic constructs derived from herpes, adeno, vaccinia, pox or reoviruses have been evaluated in several phase I trials, showing feasibility of this approach with manageable side effects as well as humoral and genomic evidence cancer-specific immune activity such as cytotoxic CD8 (+) and CD4 (+) Th1 cell infiltration; decreased VEGF and tumor-promoting miRNA levels in tumor samples and upregulated IFN-gamma and IL 12 in blood samples [132,133,134]. However, a phase II randomized trial including 103 patients with metastatic CRC failed to show a benefit of the oncolytic Pelareorep, a reovirus construct, when combined with the standard FOLFOX and bevacizumab regimen, with a shorter PFS in the investigational arm (9 vs. 7 months, HR: 1.59, *p*: 0.046), despite a higher ORR (HR: 2.52; *p*: 0.03) [135]. Recently, a phase II study investigating a more contemporary approach with dual PD-1 and CTLA-4 blockade in combination with a vaccinia virus-based oncolytic immunotherapy (Pexastimogene devacirepvec) targeting aberrant EGFR/RAS pathway was reported. The study cohort comprising 34 patients with refractory MSS metastatic CRC was split into two groups receiving single or dual checkpoint inhibition. Although there was no significant difference noted between the two regimens, there was evidence of increased peripheral cytotoxic T cell activity [136]. Similarly, Enadenotucirev, an adenovirus-based oncolytic construct was investigated in combination with nivolumab in 51 patients with refractory tumors, 35 of whom had metastatic CRC. Despite the failure to achieve a meaningful objective response, an encouraging survival rate coupled with increased intra-tumoral CD8 (+) T cell activity was observed [137]. Further improved activity was reported in a preclinical study investigating the role of herpes simplex-based oncolytic virus combined with PD-1 blockade and trametinib, a novel agent targeting the MAPK-KRAS signaling pathway in a murine model with KRAS or BRAF mutant tumors [138]. The results of ongoing trials with engineered constructs encoding multiple tumor- and immune cell-related epitopes in combination with immunomodulatory strategies are awaited with enthusiasm. 

Chimeric antigen receptor T cells (CAR-T) have revolutionized adoptive immunotherapy by sustained complete remissions achieved in patients with leukemias [139]. There exists encouraging evidence from preclinical and early phase I clinical trials evaluating the role of CAR-T or NK cells engineered to express specific immunomodulatory molecules to overcome resistance in hematologic malignancies [140,141,142]. Nevertheless, the promise of CAR-T’s has not been realized for solid tumors and several hurdles remain to be faced until meaningful clinical utility can be established. Spatial and temporal heterogeneity in tumor antigen expression, as well as obstacles to infiltrating tumor masses protected from immunologic cell kill through abnormal vasculature and a hostile TME are the major factors that pose barriers to an efficacious CAR-T cell therapy [143]. Energetic efforts have been placed to improve CAR-T activity in solid tumors by building novel cellular constructs engineered to express immunostimulatory checkpoints including CD28 and OX-40 [144], specific tumor antigens [145,146,147] or effector cytokines such as IL-12, IL-18 [148]. These strategies have yielded promising efficacy in murine models engrafted with several tumors like melanoma, mesothelioma, neuroendocrine tumors as well as CRC [149]. However, there exists evidence from phase I clinical trials showing limited activity with short-acting stable disease as the best response, highlighting the challenges that need to be addressed by future research [150]. The Keynote B79 trial, which is an ongoing study evaluating the role of an allogeneic CAR-T cell expressing an NKG2D receptor targeting both tumor cells and the TME presents a valuable example of where the field is directed. Based on previous experience suggesting preliminary efficacy with this engineered construct aiming to induce an MHC-I unrestricted immune-mediated cell kill and to antagonize immune suppressive components of the TME, the trial combines complementary immunomodulating strategies including a chemotherapy backbone with FOLFOX and pembrolizumab to drive a deeper and sustained response in a cohort of patients with refractory CRC [151].

Advances in the field of immunotherapy have led to the accumulation of an expanding pool of information on the complexity of anti-cancer immunity. Despite encouraging results investigating combinations of innovative strategies as outlined above, clinical outcomes in GI cancers have proved to pose challenges with regard to immunotherapy. In fact, preclinical and translational analyses have led to the identification of a substantial number of biomarkers, which have been implicated to play crucial roles in the generation of the immunosuppressant stroma, an inherent characteristic encountered in most GI cancers. Future studies focusing on the identification of the genomic characteristics underlying the hostile “cold” TME, as well as mechanisms elucidating the intricate interactions between effector and suppressor cell components of the tumor stroma will help to define patients who would derive benefit from these approaches. These ongoing efforts will undoubtedly deepen our understanding of the immunologic landscape and provide unprecedented insight into optimizing personalized immunotherapy for patients with GI cancers. 

## Figures and Tables

**Figure 1 vaccines-11-01545-f001:**
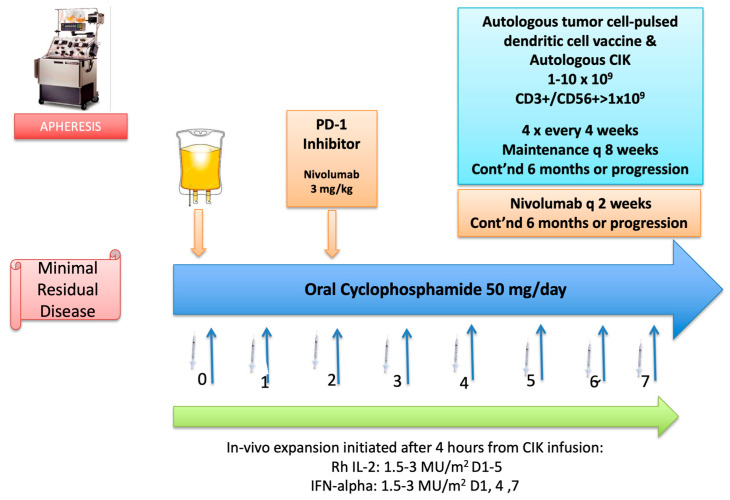
Detailed treatment protocol. The protocol was initiated following 7 days of conditioning with metronomic oral cyclophosphamide 50 mg daily. Cellular therapy was initiated on day zero (0) with DC injections and CIK infusion. DC/CIK therapy was planned to be given weekly for 4 cycles, followed by maintenance infusions every 8 weeks. Four hours after completion of the infusion, an in vivo expansion protocol with subcutaneous injections of rh-interleukin-2 (IL-2) at 3 mU/m^2^ and rh-IFN alpha (IFN-a) at 3 MU/m^2^ was initiated. IL-2 was planned to be given daily for 5 consecutive days (Days 0–4), whereas IFN-a was administered every other day as a single daily dose for 3 days (Days 0, 2, 4) after DC/CIK infusions. On day +2, nivolumab was started at a dose of 3 mg/kg given as an IV infusion over 60 min, followed by 30 min-infusions at subsequent doses every 2 weeks. The initial plan was to continue the whole protocol for 6 months and continue with maintenance DC/CIK boosters every 3 months for up to 2 years or progression; whichever occurs first. (Abbreviations: Cont’nd: continued; Rh: recombinant human).

**Figure 2 vaccines-11-01545-f002:**
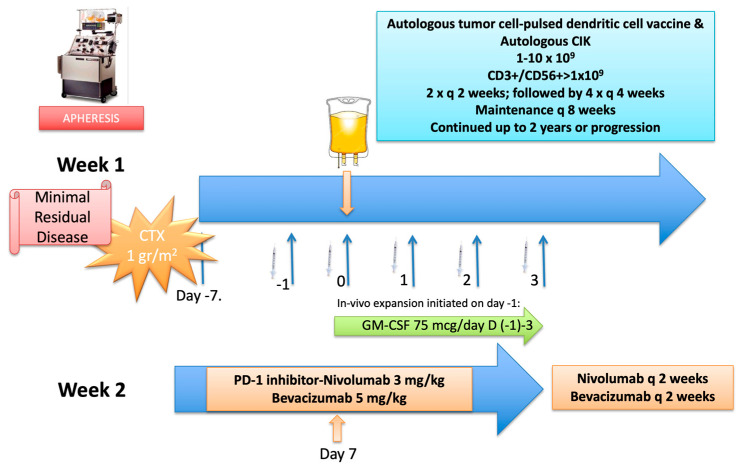
Detailed treatment protocol. The protocol was initiated on day (−7) with a conditioning regimen consisting of intravenous cyclophosphamide given as a single IV infusion at 1 gr/m^2^, followed by subcutaneous GM-SCF injections at a dose of 75 mcg/day starting on day (−1) and continued for 5 days until day 3. Cellular therapy was initiated on day zero (0) with DC injections and CIK infusion. DC/CIK therapy was planned to be given twice every 2 weeks, followed by 4 cycles given every 4 weeks and maintenance infusions every 8 weeks thereafter for up to 2 years or progression; whichever occurs first. On day 7, nivolumab was started at a dose of 3 mg/kg given as an IV infusion over 60 min, followed by 30 min-infusions at subsequent doses every 2 weeks in addition to bevacizumab at a dose of 5 mg/kg given every 2 weeks throughout the immunotherapy period. (Abbreviations: GM-CSF: granulocyte-macrophage-colony stimulating factor; CTX: cyclophosphamide).

**Figure 3 vaccines-11-01545-f003:**
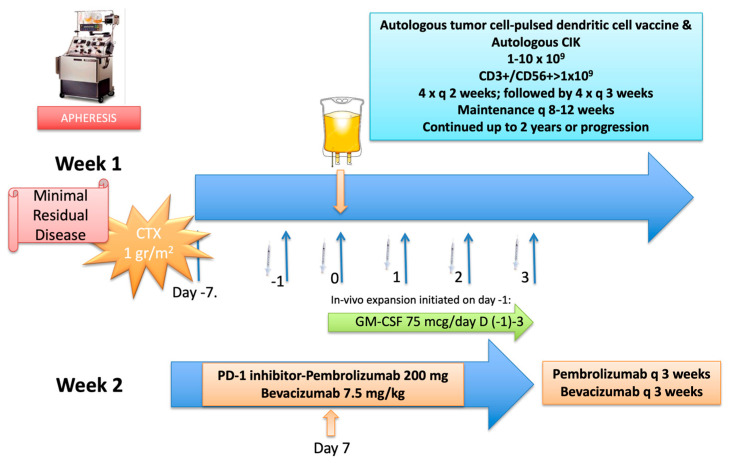
Detailed treatment protocol. The protocol was initiated on day (−7) with a conditioning regimen consisting of intravenous cyclophosphamide given as a single IV infusion at 1 gr/m^2^, followed by subcutaneous GM-SCF injections at a dose of 75 mcg/day starting on day (−1) and continued for 5 days until day 3. Cellular therapy was initiated on day zero (0) with DC injections and CIK infusion. DC/CIK therapy was planned to be given every 2 weeks for 4 cycles, followed by 4 cycles given every 4 weeks and maintenance infusions every 8 weeks thereafter for up to 2 years or progression; whichever occurs first. On day 7, pembrolizumab was started at a fixed dose of 200 mg given as an IV infusion over 60 min, followed by 30 min-infusions at subsequent doses every 2 weeks in addition to bevacizumab at a dose of 7.5 mg/kg given every 3 weeks throughout the immunotherapy period. Starting from the 3rd week after IV cyclophosphamide, oral metronomic cyclophosphamide at 50 mg daily was started and continued throughout. The second immunotherapy protocol was similar to the original protocol except that DC/CIK infusions were administered every 10 days (twice in a 3-week cycle), with omission for some DC vaccine injections and continued for 6 weeks after switching to every 3-week schedule for 6 more weeks due to limitations in cellular content obtained by apheresis. (Abbreviations: GM-CSF: granulocyte macrophage colony stimulating factor; CTX: cyclophosphamide).

**Table 1 vaccines-11-01545-t001:** Selected clinical trials with cancer vaccines in gastric and esophageal cancers.

Vaccine Type	Intervention/Neoantigen	Adjuvant/Combined Tx	Stage	Design	Outcome Measures	Status	Phase	Reference
B-cell & Monocyte	BVAC-B (Her-2)	none	IV	Her2 (+) GC, ≥1 prior lines CT 0, 4,8, 12 wks	Establish MTD, AE	completed	Ib	NCT03425773 Cellid Co, Ltd.
Allogeneic	K562-GM	Cyclophosphamide Celecoxib	NED/MRD Primary site	Esophagus & Mediastinal after standard Tx; 6 vaccines q4 wks	AE, humoral immune response	Terminated (futility)	Ib	NCT01143545 NCI
mRNA	personalized	none	IV	Esophageal, GC, CRC; ≥1 prior lines CT; 4 vaccines SC	AE, DCR, PFS, TTP, OS	Unknown	Ib	NCT03468244 Chanhai Hospital, China
B-cell epitope	Her-2	CT (Cisplatin-FU/Capecitabine)	IV	Her2 (+) GC/GE D1,14,28 vaccine vs. CT + vaccine	AE, ORR, humoral immune response	Ongoing	Ib	NCT05315830 Bengbu Med College, China
MVB-BN non-replicating viral	TAEK-VAC-Herby encoding Her-2, CD40L, TF Brachyury	Trastuzumab Pertuzumab	IV	Her2 (+) GC/GE, Breast, Chordoma 3 vaccines q3 wks	DLT	Ongoing	Ib/II	NCT04246671 Bavarian Nordic
T-cell receptor	KK-LC1 neoantigen HLA-A-01:01 restricted	2 × prior CT & IL-2 4 days & Cyclophosphamide (D-6, -5) + Fludarabine (D-6—2)	IV	KK-LC1 (+) GC, breast, lung cancer 1 vaccine infusion	DLT	Ongoing	I	NCT05035407 NCI
mRNA	PGV-002 Personalized mRNA	PD-1/PD-L1 (expansion phase)	IV	GC, esophageal, liver cancer refractory to standard CT	AE, ORR, MTD, PFS	Ongoing	I	NCT05192460 Chinese Academy of MMS NeoCura
Peptide	Personalized	Pembrolizumab, Cyclophosphamide (D-3), GM-CSF (D 1,4, 8, 15) q 3 wks	III–IV	GC/GEJ, breast, NSCLC, HCC, Merkel, GU	AE, ORR, feasibility, humoral immune response	Ongoing	I	NCT05269381 Mayo Clinic
Peptide (Da VINci)	OTSGC-A24 Multiple epitope	Nivolumab, Ipilimumab	IV	GC; refractory	AE, ORR, humoral immune response	Ongoing	Ib	NCT03784040 Natl. Uni Hospital, Singapore
Peptide	iNEO-Vac-P01 Personalized	GM-CSF	II–III maintenance	Esophageal, resectable, after resection & (neo)/adj CT+PD-1 7 vaccines D1-82	AE, RFS, OS, QoL	Ongoing	I	ZheJiang Uni; Hangzhou Neoantigen Ther Co, Ltd.
mRNA	Personalized	none	IIIC–IV	Esophageal ca; NSCLC Failure of standard Tx	AE, ORR, TTP, PFS	Ongoing	NA	NCT03908671 Zhengzhou Uni; Stemirna Ther
Adenoviral Self-replicating mRNA	GRT-C901 & R902 (predicted multiple epitopes)	Nivolumab Ipilimumab	IV	GE/G ≤1 prior CT	AE, ORR, dose finding	Ongoing	Ib	NCT03639714 Gritstone Bio, Inc.

(Abbreviations: HLA: human leucocyte antigen; Tx: treatment; CT: chemotherapy; GM-CSF: granulocyte macrophage colony stimulating factor; FU: 5-fluorouracil; NED: no evidence of disease; MRD: minimal residual disease; D: day; wks: weeks; GC: gastric cancer; CRC: colorectal cancer; NSCLC: non-small cell lung cancer; HCC: hepatocellular carcinoma; GU; genitourinary carcinoma; SC: subcutaneous; neo/adj CT: neoadjuvant/adjuvant chemotherapy; PD-1; programmed death-1; MTD: maximal toxic dose; AE: adverse effects; DCR: disease control rate; PFS: progression-free survival; RFS: relapse-free survival; TTP: time to progression; OS: overall survival; ORR; overall response rate; DLT: dose limiting toxicity; QoL: quality of life).

**Table 2 vaccines-11-01545-t002:** Selected clinical trials with cancer vaccines in colorectal cancer.

Vaccine Type	Intervention/Neoantigen	Adjuvant/Combined Tx	Stage	Design	Outome Measures	Status	Phase	Reference
Adenoviral Self-replicating mRNA	GRT-C901 & R902 (predicted multiple epitopes)	Nivolumab Ipilimumab	IV	CRC-MSS; GC/GEJ; NSCLC; GU ≤1 prior CT	AE, ORR, dose finding, Humoral immune response	Completed	Ib/II	NCT03639714 Gritstone Bio, Inc.
Adenoviral Self-replicating mRNA	GRT-C901 & R902 (predicted multiple epitopes)	Atezolizumab Ipilimumab	II–III; MRD (ctDNA (+))	CRC-MSS; Resected, after adj CT; 6 ×vaccine 2× Ipilimumab; 13× Atezolizumab q4 wks	AE, ORR (ctDNA); RFS, OS)	Terminated; reprioritization	II	NCT05456165 Gritstone Bio, Inc.
Dendritic cell	Personalized Antigen pulsed DC	mFOLFOX6	IV	Untreated mCRC; CT vs. CT + vac Vaccination in cycles 1–3; 7–9	PFS, OS, ORR	Unknown	III	NCT02503150 Second MMU, China
Adenovirus	QUILT-2.004 Ad-CEA	Avelumab CT (FOLFOX) Bevacizumab	IV	Untreated mCRC-MSS; CT (FOLFOX + Bevacizumab) vs. CT+ Avelumab + vaccine × 12	PFS	Terminated; futility on interim analysis	II	NCT03050814 NCI
Viral (modified vaccinia Ankara-Bavarian Nordic)	CV 301 MVA-BN CEA/MUC1 & Fowlpox booster	Bifunctional fusion protein composed of IgG1 PD-L1 & TGF-beta; IL-15 fusion protein IL-12	IV	mCRC; small bowel ca. ≥2 prior lines CT Triple vs. quadruple Tx (±IL12)	AE, ORR; PFS; OS	Completed	II	NCT04491955 NCI
Autologous	Cryovax; Personalized autologous tumor	Bioengineered allogenic immune cells (AlloStim)	IV	>2 previous lines of CT; Vaccine ×6 over 10 wks	DLT, QoL, Humoral immune response; ORR	Completed	II	NCT02380443 Immunovative Therapies Ltd.
Allogenic engineered	GVAX whole tumor cell engineered to secrete GM-CSF	Cyc Guadecitabine	IV	mCRC; stable on 1-2nd Line CT; Vaccine × 1 q 4 wks	AE; TIL; PFS	Completed	I	NCT01966289 Sydney Kimmel Cancer Center
Peptide	PolyPEPI1018 Six synthetic CTA neoantigens	TAS 102	IV	mCRC-MSS; ≤2 prior CT Vaccine q2 weeks × 7 doses	AE, PFS, ORR; OS	Completed	I	NCT05130060 Mayo clinic
mRNA	mRNA 5671/V941-001	Pembrolizumab	IV	mCRC- KRASm-MSS; NSCLC; Pancreas Vaccine q 3 wks 9 doses vs. Vaccine + Pembrolizumab	AE, DLT	Completed	I	NCT03948763 Merck Sharp & Dohme LLC
Synthetic peptide	Her-2 & CEA HLA A2/A3 restricted	GM-CSF Tetanus toxioid Montanide ISA 51	II-IV	CRC	AE; T cell response in lymph node	Terminated Slow accrual	I	NCT00091286 University of Virginia
Alphavirus replicon particles	AVX 701 CEA	none	III	CRC; After adjuvant CT Vaccine ×4 q 3 wks	AE; Humoral immune response	Completed	I	NCT01890213 Duke University
Inactivated virus	Influenza vaccine	none	Early, operable	CRC; intratumoral injection × 1	AE, local immune responses	Completed	II	NCT04591379 Zealand University
Plasmid DNA	MYPHISMO Tet-MYB	BGB-A317 (Anti-PD-1 IgG4 mab)	IV	mCRC, ACC; refractory; Vaccine × 6 q 7d	AE, ORR; CBR, PFS	Completed	I	NCT03287427 Peter Mc Callum Cancer Centre, Au
Adenoviral Self-replicating mRNA	GRANITE-GRT-C901 & R902 (predicted multiple epitopes)	Atezolizumab Ipilimumab 5-FU Bevacizumab	IV	mCRC; maintenance following 1st L CT (FOLFOX-Bevacizumab)	Molecular response; PFS	Ongoing	II/III	NCT05141721 Gritstone Bio; Inc.
Liposomal	StimVax; MUC-1	AlloStim; Allogenic immune cells	IV	mCRC-MSS; >2 previous lines of CT; Vaccine 3 cycles (×5/wk q 6 wks)	AE, OS	Ongoing	IIb	NCT04444622 Immunovative Ther, Ltd.
Dendritic cell	Personalized neoantigen	Nivolumab	IV; MRD	mCRC; resected liver met; HCC Vaccine × 10 doses q 2 wks	RFS, Humoral immune response	Ongoing	II	NCT04912765 Natl Cancer Centre, Singapore
Peptide	Personalized synthetic neoantigen	Pembrolizumab Imoquimod Sotigolimab		mCRC; any line; Pancreas ca; Vaccine × 7-11 dose q 2 wks	AE, Feasibility, ORR; PFS; OS	Ongoing	I	NCT02600949 MD Anderson Cancer Center
Chimeric recombinant protein; Recombinant viral	KISIMA-01; ATP-128-Three neoantigens+ TLR agonist; VSV-GP128-booster	Ezabenlimab (anti-PD-1)	IV	mCRC-MSS; 1st Line CT; refractory; liver-only	AE, PFS; ORR; MTD; RFS	Ongoing	I/II	NCT04046445 Amal Ther.
Peptide	KRAS peptide	Nivolumab Ipilimumab	IV	mCRC; pancreas; >2 previous lines of CT; Vaccine × 3 q 7 d; × 5 boosters q 8 wks	AE; humoral immune response; DFS; ORR; PFS; OS	Ongoing	I/II	NCT04117087 Sydney Kimmel Cancer Center
Peptide	PolyPEPI1018 7 peptide neoantigens; Montanide	Atezolizumab	IV	mCRC-MSS >2 previous lines of CT;	AE, ORR; PFS; OS; Humoral immune response	Ongoing	II	NCT05243862 Treos Bio Ltd.
Yeast cell particles	PalloV-CRC; Allogenic tumor cells delivered on yeast cell particles	none	I-IV	CRC prior to surgery Vaccine × 4 q 4 wks	AE; humoral immune response	Ongoing	I	NCT03827967 Cancer Insight, LLC
Dendritic cell	COREVAX-1; Dendritic cells pulsed with autologous tumor cells	IL-2 (D3-7)	IV; MRD	mCRC; Following resection	AE; humoral immune response; RFS; OS	Ongoing	II	NCT02919644 Instituto Scientifico Romagnolo per lo Studio e la cura dei Tumori
Peptide	CLAUDE; EO2040; TAA-CD8/CD4 T cell epitopes	Montanide Nivolumab	II-IV; MRD (ct-DNA (+)	mCRC; Following resection & standard Tx	ORR at 6 months; AE; DFS; OS	Ongoing	II	NCT05350501 Enterome
Lipid conjugated oligonucleotide & peptide	ELI-002; KRAS/NRAS	none	MRD; ct-DNA (+)	mCRC; pancreatic; NSCLC; Vaccine × 4 q 7 d; × 8 boosters/4 wks	Ct-DNA clearance; RFS; OS	Ongoing	I/II	NCT05726864 Elicio Ther
Adenovirus	Tri-AD5 Trivalent CEA/MUC-1/Brachyury	nogapendekin alfa inbakicept (IL-15 agonist-fusion protein)	Prevention	Lynch Syndrome; colon adenomas & CRC NED; Vaccine × 4 (wks 0, 4, 8, 52)	Cumulative incidence of adenomas; extracolonic carcinomas; humoral immune responses	Ongoing	IIb	NCT05419011 NCI
Adenovirus	Nous-209; Gad-209-FSP priming; MVA-209-FSP booster	None	Prevention	gLynch Syndrome; NED after non-sporadic MMRd malignant tumors; Vaccine D1/booster wk 8	AE; humoral immune responses; response in colorectal adenomas; incidence of Lynch-associated carcinomas	Ongoing	I/II	NCT05078866 NCI

(Abbreviations: DC: dendritic cell; CEA: carcinoembryonic antigen; MUC: mucin; KRAS: kirsten rat sarcoma virus; NRAS: neuroblastoma ras viral oncogene homolog; Tx: treatment; CT: chemotherapy; GM-CSF: granulocyte macrophage colony stimulating factor; FOLFOX: oxaliplatin-5-fluorouracil infusion-leucovorin; wk: week; d: days; IL: interleukin; NED: no evidence of disease; MRD: minimal residual disease; ct-DNA (+): circulating tumor DNA; wks: weeks; GC: gastric cancer; GEJ: cancer involving distal osephageal; cardia; proximal gastric areas; CRC: colorectal cancer; mCRC: metastatic CRC; CRC-MSS: microsatellite stable CRC; NSCLC: non-small cell lung cancer; HCC: hepatocellular carcinoma; GU; genitourinary carcinoma; gLynch: germline Lynch; MMRd: mismatch repair deficient; SC: subcutaneous; neo/adj CT: neoadjuvant/adjuvant chemotherapy; PD-L1; programmed death ligand-1; MTD: maximal toxic dose; AE: adverse effects; CBR: clinical benefit rate; DCR: disease control rate; PFS: progression-free survival; RFS: relapse-free survival; TTP: time to progression; OS: overall survival; ORR; overall response rate; DLT: dose limiting toxicity; QoL: quality of life).

## Data Availability

Data is not available due to ethical restrictions from the Ministry of Health.

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
