# Peer review of "Clinical Applications of Combined Immunotherapy Approaches in Gastrointestinal Cancer: A Case-Based Review"

_vaccines, 2023, doi:10.3390/vaccines11101545_

Round 1

Reviewer 1 Report

Eralp & Ates present here a comprehensive narrative review about combined immunotherapy approaches in GIT cancers and provide a relevant case series including 3 patients.

I would like to comment the authors for their great effort to put together an immuno-oncology review and provide the rationale behind the immunotherapy approaches in solid malignancies, and in particular GIT cancers.

In addition, the 3 patients explicitly presented are of unique importance and provide real world evidence for the application of combined immunotherapy; an approach that we commonly see at clinical trials (and not in clinical practice). More and more similar case presentations should be presented to familiarize clinicians with these novel applications.

The manuscript is well written, and I hardly have any comments for its improvement.

I would advise the authors to revisit the draft for minor English corrections that were seldomly found throughout the manuscript. Try also to revise the connection of sentences in introduction lines 30-34, which is vague.

One other point to consider is moving the methods in a separate section, before cases presentation (is minor, even if you leave it as it is, is fine).

In patient 3, state at the beginning that is a male.

I would rather see the "CTLA" target termed as "CTLA-4" throughout the manuscript, since, to my knowledge, there is no other target in the CTLA family of receptors.

Also, did you have PD-L1 status in patient 1&2? If yes, please provide it.

Lastly, if I miss one intellectual point is whether were there any biomarkers applied for selection of patients in the respective trials, and if you applied those in your patients. If not, you may want to address the lack of biomarkers for selection of patients as a point of discussion, in your otherwise strong conclusion/discussion section.

Eralp & Ates present here a comprehensive narrative review about combined immunotherapy approaches in GIT cancers and provide a relevant case series including 3 patients.

I would like to comment the authors for their great effort to put together an immuno-oncology review and provide the rationale behind the immunotherapy approaches in solid malignancies, and in particular GIT cancers.

In addition, the 3 patients explicitly presented are of unique importance and provide real world evidence for the application of combined immunotherapy; an approach that we commonly see at clinical trials (and not in clinical practice). More and more similar case presentations should be presented to familiarize clinicians with these novel applications.

The manuscript is well written, and I hardly have any comments for its improvement.

I would advise the authors to revisit the draft for minor English corrections that were seldomly found throughout the manuscript. Try also to revise the connection of sentences in introduction lines 30-34, which is vague.

One other point to consider is moving the methods in a separate section, before cases presentation (is minor, even if you leave it as it is, is fine).

In patient 3, state at the beginning that is a male.

I would rather see the "CTLA" target termed as "CTLA-4" throughout the manuscript, since, to my knowledge, there is no other target in the CTLA family of receptors.

Also, did you have PD-L1 status in patient 1&2? If yes, please provide it.

Lastly, if I miss one intellectual point is whether were there any biomarkers applied for selection of patients in the respective trials, and if you applied those in your patients. If not, you may want to address the lack of biomarkers for selection of patients as a point of discussion, in your otherwise strong conclusion/discussion section.

Author Response

Please see file as attached;

Reviewer 2 Report

In this review, Eralp et al. put forth the proposition that gastrointestinal (GI) tract cancers are common and often fatal, with limited treatment success, even when using targeted therapy. The review also discusses how immunotherapy, which includes immune checkpoint inhibitors, displays promise but demonstrates its effectiveness primarily within a subgroup characterized by microsatellite instability. To enhance treatment outcomes, researchers are actively investigating diverse immunotherapy approaches, such as cancer vaccines and adoptive cell therapies. Within this context, the review delves into personalized dendritic cell cancer vaccines and cytokine-induced killer cell therapy, highlighting their combined immunotherapy strategies as a potential path forward in managing GI cancer.

#I have observed that the topic exhibits originality and relevance within the field, effectively addressing a specific gap in current research. I firmly believe that this review would be highly valuable in the context of gastrointestinal (GI) research, particularly in its exploration of diverse immunotherapy methods, encompassing cancer vaccines and adoptive cell therapies.

#The methodology is fine and no further control is required.

#I found the conclusion to be in line with the evidence and arguments presented.

#The references are well-updated.

#The figures and tables are sufficient.

Thus, I recommend accepting it in its current form.

Minor editing of the English language is required.

Author Response

September 24, 2023

Re: Response to comments on manuscript

Manuscript ID: vaccines-2617166;

Title “Clinical applications of combined immunotherapy approaches in gastrointestinal cancer: a case-based review”

            Dear Sir,

Thank you for the opportunity to review our work titled “Clinical applications of combined immunotherapy approaches in gastrointestinal cancer: a case-based review”. I appreciate the careful review and constructive suggestions. I would also like to express my sincere gratitude for your valuable comments and recommendations. I am very pleased to hear that our work has been evaluated as well written and providing valuable contribution to the field of GI immunotherapy research. I have read your comments carefully and made the necessary revisions on language. The article has been proofread and all spelling mistakes and punctuation errors were corrected. Some sentences were modified.

I believe that the manuscript is substantially improved after making the suggested edits.

Thank you for your time and consideration to evaluate our manuscript.

Sincerely yours,

Yesim ERALP, MD

Professor of Medical Oncology

Acıbadem University, Istanbul TURKEY

Reviewer 3 Report

In current review article, Eralp and Ates report their experience with personalized dendritic cell cancer vaccine and cytokine-induced killer cell therapy in three patients with GI cancers and summarize their current clinical data on combined immunotherapy strategies. In current article, in addition to case report, author has comprehensively described about the cancer immunology and immunotherapy and immunotherapy in GI cancers and their cellular treatment systems. Author also summarized the selected clinical trials with cancer vaccines in gastric and esophageal cancers and cancer vaccines in colorectal cancer. This current review article will be critical for the researcher working in the field of cancer vaccine and combined immunotherapy approaches in GI and other cancers. This reviewer recommends this review article for the publication in vaccines.

Author Response

September 24, 2023

Re: Response to comments on manuscript

Manuscript ID: vaccines-2617166;

Title “Clinical applications of combined immunotherapy approaches in gastrointestinal cancer: a case-based review”

            Dear Sir,

Thank you for the opportunity to review our work titled “Clinical applications of combined immunotherapy approaches in gastrointestinal cancer: a case-based review”. I appreciate the careful review and would like to express my sincere gratitude for your valuable comments. I am very pleased to hear that our work has been evaluated as well written and providing valuable contribution to the field of GI immunotherapy research.

Thank you for your time and consideration to evaluate our manuscript.

Sincerely yours,

Yesim ERALP, MD

Professor of Medical Oncology

Acıbadem University, Istanbul TURKEY